# Glioblastoma Metabolism: Insights and Therapeutic Strategies

**DOI:** 10.3390/ijms24119137

**Published:** 2023-05-23

**Authors:** Chloé Bernhard, Damien Reita, Sophie Martin, Natacha Entz-Werle, Monique Dontenwill

**Affiliations:** 1UMR CNRS 7021, Laboratory Bioimaging and Pathologies, Tumoral Signaling and Therapeutic Targets, Faculty of Pharmacy, University of Strasbourg, 67405 lllkirch, France; chloe.bernhard@etu.unistra.fr (C.B.); damien.reita@chru-strasbourg.fr (D.R.); sophie.martin@unistra.fr (S.M.); natacha.entz-werle@chru-strasbourg.fr (N.E.-W.); 2Laboratory of Biochemistry and Molecular Biology, Department of Cancer Molecular Genetics, University Hospital of Strasbourg, 67200 Strasbourg, France; 3Pediatric Onco-Hematology Unit, University Hospital of Strasbourg, 67098 Strasbourg, France

**Keywords:** glioblastoma, metabolism, cancer stem cells, tumorigenic processes, therapy

## Abstract

Tumor metabolism is emerging as a potential target for cancer therapies. This new approach holds particular promise for the treatment of glioblastoma, a highly lethal brain tumor that is resistant to conventional treatments, for which improving therapeutic strategies is a major challenge. The presence of glioma stem cells is a critical factor in therapy resistance, thus making it essential to eliminate these cells for the long-term survival of cancer patients. Recent advancements in our understanding of cancer metabolism have shown that glioblastoma metabolism is highly heterogeneous, and that cancer stem cells exhibit specific metabolic traits that support their unique functionality. The objective of this review is to examine the metabolic changes in glioblastoma and investigate the role of specific metabolic processes in tumorigenesis, as well as associated therapeutic approaches, with a particular focus on glioma stem cell populations.

## 1. Introduction

Glioblastomas (GB) are highly malignant and among the most challenging cancers. The standard of care for the treatment of GB is still the Stupp protocol, which was first introduced in 2005. This protocol combines surgery, radiotherapy, and chemotherapy with temozolomide (TMZ) [1]. Although this treatment approach has been widely used, it fails to achieve long-term remission for patients, due in part to the molecular heterogeneity and plasticity of GB cells, including GB stem cells (GSCs). GB are characterized by a high degree of heterogeneity, whether in terms of their histological and molecular characteristics, cellular origin, topography, growth patterns, and sensitivity to therapy. This heterogeneity exists not only between different tumors (intertumoral heterogeneity), but also within the same tumor (intratumoral heterogeneity) [2,3]. In 2010, a large-scale transcriptomic analysis of GB revealed the presence of molecular signatures that define different clinically relevant subtypes of adult GB, including classical, mesenchymal, and proneural subtypes [4,5]. Later, the Neftel classification described the heterogeneity of different cellular states within a single tumor, distinguishing four major states (neural progenitor-like, oligodendrocyte progenitor-like, astrocyte-like, and mesenchymal-like) that are modulated by the microenvironment and favored by distinct genetic alterations. The proportion of cells in each state within a tumor can vary over the course of tumor evolution and under the pressure of treatments [6]. The search for new therapies for GB patients has predominantly centered around molecular targets discovered through comprehensive genomic analyses such as The Cancer Genome Atlas (TCGA). These targets are recurrent molecular alterations that disrupt important pathways that regulate growth, cell cycle, autophagy, DNA repair, apoptosis, angiogenesis, and immune checkpoints and are considered as potential therapeutic targets for GB treatment. Despite these efforts, no significant clinical improvement was achieved, probably due to the significant intratumoral heterogeneity [7]. The interest in tumor metabolism has risen over the past decade. Indeed, targeting the tumor metabolism has proven promising, particularly for the elimination of GSCs [8]. Furthermore, combining conventional treatments with therapies targeting tumor metabolism may result in better care for GB patients, with lower doses of chemotherapy and fewer side effects, while still achieving effective anticancer results and potentially overcoming treatment resistance [9]. The understanding of protumorigenic metabolic mechanisms and pathways is necessary to define relevant therapeutic targets and subsequently to propose effective combination therapies. Advancements in research and technology have led to the identification of key differences between tissues and tumors that could be targeted in cancer therapy. However, research has also revealed the heterogeneity, complexity, and plasticity of tumor metabolism in both preclinical and clinical models [10].

## 2. Tumoral Metabolic Reprogramming

Reprogramming of metabolism in cancer is a multifaceted process driven by genetic and environmental factors. Tumoral metabolic reprogramming provides energy (ATP), precursors for macromolecules (carbohydrates, lipids, proteins, and nucleic acids), and reducing equivalents essential for the extensive proliferative activity of cancer cells. Alterations in glycolysis, Oxidative Phosphorylation (OXPHOS), the Pentose Phosphate Pathway (PPP), lipids, amino acids and nucleotides metabolism have been observed (Figure 1) [11,12]. Moreover, cancer cells adapt to their environment by reprogramming their cellular metabolism, which allows them to grow, survive and proliferate in a constantly changing environment [13].

GB cells primarily rely on glucose as their metabolic fuel, but they can also utilize amino acids such as glutamine and glutamate, lipids such as fatty acids and lipid droplets [14], and other sources such as acetate [15], depending on the genetic background and the tumor microenvironment. For example, GB cells expressing constitutively active AKT are totally glucose dependent [16]. Glucose can either undergo aerobic glycolysis leading to lactate formation or be oxidized in the mitochondria through the Tricarboxylic Acid Cycle (TCA), both of which are energy-producing pathways. The TCA cycle provides reducing equivalents that supply the function of the respiratory chain, the site of massive ATP energy production. In contrast, the conversion of pyruvate into lactate (aerobic glycolysis) yields only a small amount of energy. Beyond energy needs, cancer cells need to generate biomass. Thus, in GB, metabolic intermediates of glycolysis are shunted to the Pentose Phosphate Pathway but also to the serine and lipid biosynthetic pathways. The upregulation of the Pentose Phosphate Pathway stimulates the nucleotide synthesis which is essential for DNA replication and repair, but also to produce reducing equivalents (NADPH) supporting redox homeostasis and lipid biosynthesis [17]. The TCA cycle, supplied with glucose and alternative fuels such as glutamate, fatty acids, acetate, and ketones, provides energy intermediates and anabolic precursors [15,18,19].

**Figure 1 ijms-24-09137-f001:**
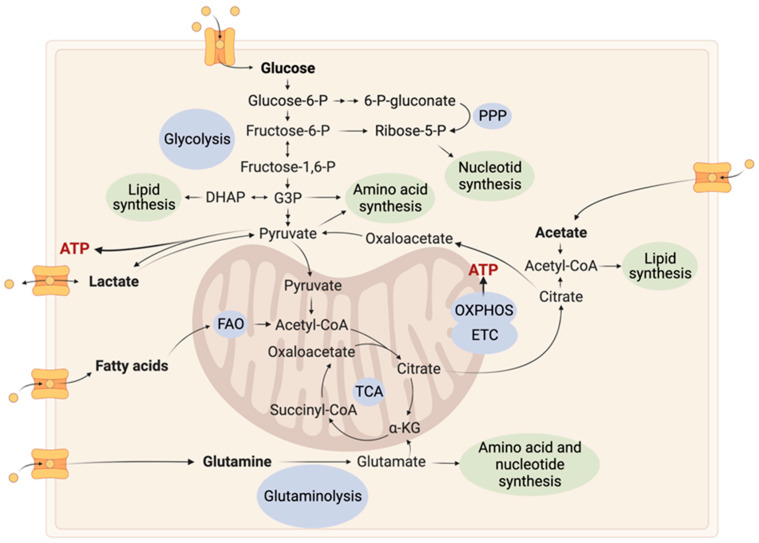
Active metabolic pathways in glioblastoma. Cancer cells exhibit a variety of metabolic changes, including increased glycolysis leading to lactate production and glutaminolysis, which provides energy for the TCA cycle. Metabolic nutrients are highlighted in bold. Metabolic intermediates are channeled into nucleic acid, amino acid, and lipid biosynthetic pathways. These metabolic pathways are intricately connected [10,18]. α-KG: α-Ketoglutarate; ATP: Adenosine Triphosphate; DHAP: Dihydroxyacetone Phosphate; ETC: Electron Transport Chain; FAO: Fatty Acid Oxidation; G3P: Glyceraldehyde-3-Phosphate; OXPHOS: Oxidative Phosphorylation; P: Phosphate; PPP: Pentose Phosphate Pathway; TCA: Tricarboxylic Acid Cycle. Figure created with BioRender.com.

The metabolic changes in GB are attributed to mutations in tumor-suppressor genes and oncogenes, as well as the influence of the surrounding microenvironment. The relationship between metabolic and molecular reprogramming is complex and interdependent [20,21]. Oncogenic signaling pathways (such as PI3K-AKT-mTOR, MYC, and Ras) or the inactivation of tumor-suppressor genes (such as TP53) have been demonstrated to modulate tumor metabolism [22]. The PI3K-AKT-mTOR signaling pathway plays a crucial role in regulating tumor metabolism, including promoting glycolysis and anabolic metabolism of nucleotides, lipids, and proteins, and is frequently aberrantly activated in GB. mTOR regulates cell growth, translation, and the initiation of autophagy [23]. Hypoxia and hypoxia-inducible factors (HIFs) also play a crucial role in promoting metabolic reprogramming in cancer cells as well as in stemness and therapy resistance [24,25].

Since metabolism plays a pivotal role in GB tumorigenesis and progression, it presents a valuable avenue for exploring potential therapeutic approaches in treating GB.

### 2.1. Glucose Metabolism

The Warburg effect, also known as aerobic glycolysis, is a metabolic feature commonly found in cancer cells. This metabolic change is characterized by increased glucose uptake and consumption, resulting in an increase in lactate production, even in the presence of oxygen [10]. Aerobic glycolysis produces less energy than oxidative metabolism, but allows cancer cells to rapidly convert available resources into biomass (lipids, nucleotides, and amino acids) [17]. This is also critical for the survival and proliferation of cancer cells under hypoxic conditions [20,26]. Lactate production is accompanied by the production of protons, both of which pass through the extracellular environment via monocarboxylate (MCT 1,4) and Na^+^/H^+^ type 1 (NHE1) transporters, respectively. This leads to the acidification of the microenvironment and the development of an immunosuppressive environment, promoting the growth and invasion of cancer cells [27,28]. Lactate can also trigger angiogenesis by being imported into endothelial cells [20]. The Warburg effect results from an increase in the transcription of genes coding for Glucose Transporters (GLUT) and glycolysis-associated enzymes, making them potential targets for therapy (Figure 2) [29].

The use of 2-Deoxyglucose (2-DG) has shown potential as a cytotoxic agent in cancer treatment due to its ability to inhibit glycolysis and disrupt cellular metabolism. A phase II clinical trial found that 2-DG was both safe and well tolerated in GB patients [30,31]. However, its short half-life and adverse effects limit its therapeutic potential. To address these issues, novel analogues and prodrugs of 2-DG have been developed. One such analogue, WP1122, has demonstrated promising results in preclinical studies. WP1122 releases 2-DG and has a longer half-life, good oral bioavailability and is well tolerated in animal models. WP1122 has completed a phase I clinical trial (NCT05195723) and is planned to enter in clinical trial for the treatment of GB patients [32]. The future perspectives for 2-DG and its analogues in anticancer therapy lie in their potential synergistic effects when combined with other potent cytotoxic agents.

Other potential targets are GLUT1 and GLUT3 isoforms, which are commonly overexpressed in GB [33]. GLUT1 isoform is crucial for glucose transport across the blood–brain barrier, and GLUT3 isoform is mainly expressed in neurons and overexpressed in GSCs and has a higher affinity for glucose and has been linked to poor prognosis in GB patients [34,35]. Silencing of *GLUT3* in GSCs [35], as well as *GLUT1* silencing or pharmacological inhibition of GLUT1 using WZB117 [36], reduced tumor formation in vivo. Metabolic dependency on GLUT3 has been observed in some classical and proneural GB subtypes, driven by an abnormal αvβ3 integrin expression [37]. Moreover, GLUT3 expression has been linked to bevacizumab resistance [38]. A recent study revealed that histone deacetylase 2 knockdown resulted in the suppression of GLUT3 expression by upregulating miR-3189, leading to an antitumorigenic effect [39]. Currenly, there are few GLUT inhibitors and no GLUT3 specific inhibitors, and their efficacy in GB and potential toxicity to normal cells have not been extensively studied [33].

Another therapeutic approach involves targeting glycolytic-associated enzymes. Many glycolytic enzymes are upregulated in cancer cells, and some have been identified as crucial for tumor growth, such as Hexokinase 2 (HK2), aldolase A (ALDOA), pyruvate dehydrogenase kinase 1 (PDK1), Pyruvate Kinase M2 (PKM2), 6-phosphofructo-2-kinase/fructose-2,6-biphosphatase 4 (PFKFB4) and Enolase 1 and 2 (ENO1 and ENO2) [40,41]. HK2 plays a significant role in the inhibition of mitochondria-mediated apoptosis and has been linked to tumor grade and poor prognosis in GB [42]. Depletion of HK2 in GB xenograft models decreased tumor proliferation and angiogenesis but increased tumor invasion [42]. The inhibition of HK2 in cancer therapy may be doubted due to its non-specificity and systemic toxicity, as well as the fact that tumors expressing HK2 can also express HK1 [9]. However, ketoconazole and posaconazole, members of the azole class of antifungals, have been identified as inhibitors of tumor metabolism that target the HK2-associated gene signature. This treatment has been shown to induce apoptotic cell death in vitro and reduce tumor growth in vivo in GB models [43]. These promising results have led to the initiation of phase I clinical trials (NCT04869449 and NCT04825275) in patients with brain tumors. PFKFB4 has been identified as another critical enzyme for the maintenance of GSCs, and its high expression is correlated with poor prognosis in GB patients [40]. The final step in glycolysis is catalyzed by Pyruvate Kinase (PK). Normally, the PKM2 isoform is only expressed in embryonic tissues and adult stem cells. In cancer, however, PKM2 is overexpressed, leading to a reprogramming of glucose utilization. PKM2 has a lower affinity for phosphoenolpyruvate (PEP) compared to PKM1, resulting in low enzymatic activity, allowing the metabolic intermediates produced upstream to be used for biosynthetic processes [44]. PKM2 has the ability to directly bind to histone H3 and induce its phosphorylation, thereby enabling the expression of genes such as *CCND1* (Cyclin D1) and *MYC*. There is a correlation between the phosphorylation levels of histone H3, the levels of nuclear PKM2, the grade of malignancy, and the patient’s prognosis [45]. Additionally, under oxidative stress, PKM2 can be translocated into mitochondria and prevent apoptosis by phosphorylating Bcl-2 [46]. Recently, TP-1454, a compound that activates PKM2, entered a phase I clinical trial and is currently being evaluated in combination therapy for patients with solid progressive tumors (NCT04328740). A study has shown that trametinib (a MEKK inhibitor) specifically targets the PKM2/c-MYC pathway, leading to the suppression of glycolysis and growth in glioma cells. A phase II clinical trial (NCT03919071) is currently recruiting newly diagnosed high-grade glioma patients who have undergone radiation therapy to evaluate the effectiveness of a combined treatment of trametinib and dabrafenib (a MAPK inhibitor) [47]. Enolase is another crucial enzyme, catalyzing the formation of PEP. Enolase activity is determined by three genes: *ENO1*, ubiquitously expressed; *ENO2*, expressed only in neural tissue; and *ENO3*, expressed in muscle tissue. An analysis of TCGA data revealed that some GB exhibit homozygous deletions in *ENO1*, resulting in the abnormal expression of ENO2. This highlights a metabolic vulnerability of GB cells lacking *ENO1*, in which *ENO2* deletion has inhibited the growth, survival and tumorigenic properties of cancer cells [48]. PDK1 is another promising target for anticancer therapy, as it has been demonstrated to play a critical role in the survival of GSCs. PDK1 functions as a negative regulator of pyruvate dehydrogenase (PDH), reducing the supply of acetyl-CoA in the mitochondria and thus leading to a decrease in oxidative mitochondrial metabolism [49]. Dichloroacetate (DCA), usually used to treat lactic acidosis, has been shown to reverse the Warburg effect by inhibiting PDK1. This activates mitochondrial PDH and thus increases the flow of pyruvate into the mitochondria, leading to membrane depolarization, an increase in mitochondrial reactive oxygen species (ROS) levels, and ultimately the induction of apoptosis. DCA is particularly noteworthy in its ability to enhance radiation sensitivity. Indeed, in animal models of GB, DCA has been shown to improve survival when combined with radiotherapy [50]. Although phase I (NCT01111097) and phase II (NCT00540176) clinical trials have been conducted, clinical data on its efficiency remain lacking. However, preliminary results have shown that this drug crosses the blood–brain barrier and is well tolerated by patients.

Lactate Dehydrogenase (LDH) is a major enzyme that catalyzes the interconversion of pyruvate to lactate. There are two types of LDH found in cancers: LDH-A, involved in the Warburg effect, catalyzes the conversion of pyruvate to lactate; and LDH-B catalyzes the reverse reaction, i.e., the conversion of lactate to pyruvate. LDH-A is overexpressed in GB, particularly in hypoxic and invasive areas. Overexpression of LDHA has been linked to tumor initiation, maintenance and progression, as well as poor prognosis in many types of cancer, including decreased survival in GB patients treated with radiotherapy [51,52]. In vitro studies have shown that LDH-A inhibitors, such as NHI-1 and NHI-2, affect the maintenance of GSCs, inhibit cell differentiation and induce apoptosis processes [53]. More recently, research has demonstrated that lactate fuels GB anaplerosis by replenishing the TCA cycle in absence of glucose through LDH-B activity [54]. A metabolic cooperation between glycolytic and oxidative cancer cells has also been described. Cells under hypoxic conditions overexpress LDH-A, exhibiting glycolytic phenotype that produce large amounts of lactate and secretes it into the microenvironment. More distant oxidative cells that overexpress LDH-B uptake lactate and convert it to pyruvate to fuel the TCA cycle. The combined ablation of both LDH isoforms, but not just one, has been shown to decreased tumor growth, improved mouse survival, and increased sensitivity to radiotherapy. In addition, the anti-epileptic drug stiripentol has been shown to inhibit the activity of both LDH isoforms and has been proven to effectively decrease the growth of GB [55]. Another promising compound in this regard is gossypol, a natural polyphenolic drug, which targets the Bcl-2 protein family and various dehydrogenases, including aldehyde dehydrogenases (ALDH) and LDH-A. Gossypol has been shown to inhibit the growth and induce cell death of TMZ-resistant GB cells, including GSC, with increased sensitivity [56]. Clinical trials evaluating gossypol for the treatment of GB have been conducted. Different patient responses have been observed, including tumor stability for over seven months of treatment (phase I: NCT00390403; phase II: NCT00540722) [57]. One therapeutic approach for inhibiting glycolysis involves the use of 5-Aminolevulinic Acid (5-ALA), a natural heme precursor that accumulates in GB tumors to a greater extent than in healthy cells. 5-Aminolevulinic Acid has been shown that 5-ALA can impair glycolysis by inhibiting LDH, leading to cell death. The use of 5-ALA as an adjuvant for visualization of high-grade gliomas was approved by the Food and Drug Administration (FDA) in 2017 and has since become a common tool for guiding brain cancer resection. 5-ALA’s potential as a cancer treatment could improve the therapeutic care for GB patients [58]. Melatonin has been demonstrated to impact glycolysis in GSCs leading to cell death by repressing LDH-A and MCT4 expression, resulting in reduced lactate production and decreased intracellular pH and ATP levels. Additionally, this approach caused an increase in ROS and blockage of cell cycle progression, ultimately resulting in cell death [59]. Recently, researchers have developed a novel approach to target GB cells by using therapeutic nanoparticles. These nanoparticles are designed to penetrate the blood–brain barrier and specifically recognize the membrane components of GB cells. Upon entering the tumor, lactate oxidase within the nanoparticles converts lactate into pyruvic acid and hydrogen peroxide. Pyruvic acid blocks histone expression and induces cell cycle arrest, while hydrogen peroxide generates singlet oxygen to kill GB cells through a reaction with a delivered photosensitizer [60]. Recent studies have demonstrated the potential of oxaloacetate, a TCA cycle metabolite, to suppress the Warburg effect in GB [61]. Treatment with oxaloacetate resulted in reduced tumor growth and increased survival in animal models of implanted GB [62]. These results led to the initiation of a clinical trial with Anhydrous Enol-Oxaloacetate (AEO) oral administration in combination with standard therapy for newly diagnosed GB patients (NCT04450160).

In summary, the Warburg effect, lactate production, overexpression of Glucose Transporters and glycolysis-associated enzymes are important features of cancer cell metabolism that offer potential targets for therapy, particularly in certain GB subtypes with specific metabolic vulnerabilities.

### 2.2. Amino Acid Metabolism

#### 2.2.1. Glutamine Metabolism

Reprogrammed glutamine metabolism is essential for the survival and proliferation of cancer cells, providing TCA cycle intermediates, nucleotides, amino acids, and fatty acids biosynthesis, regulating redox homeostasis, and modulating α-KG levels involved in DNA and histone demethylation. Reprogrammed glutamine metabolism stimulates the synthesis of glutathione, which helps fight against oxidative stress [63] and was correlated to an increased capacity of resistance to radio- and chemo-therapies [64]. As a result, targeting glutamine metabolism has emerged as a promising therapeutic strategy in many cancers, including GB (Figure 3).

Glutamine enters cells via the SLC1A5/ASCT2 transporters and is converted to glutamate within the mitochondria via an oxidative deamination reaction catalyzed by glutaminases (GLS). This reaction converts glutamine to glutamate and ammonia. Glutamate is then converted to α-KG by glutamate dehydrogenase (GDH), replenishing TCA cycle intermediates [65]. Glutamine can also be converted into pyruvate by the malic enzyme to fuel TCA cycle or produce lactate. GB cells are able to use microenvironment-derived glutamate to generate glutamine through the expression of glutamine synthetase [15,19]. Regarding circulating glutamine, it does not represent a major fuel for the TCA cycle. Low expression of glutamine synthetase has been linked to improved prognosis in GB patients, with an average survival time that is two-fold longer [66]. Overexpression of this enzyme has been found in GSC populations [67]. Some cancer cells are extremely sensitive to glutamine deprivation [68], while other cells are insensitive due to autonomous glutamine production via glutamine synthetase expression [67].

Glutaminase inhibition (GLS) as a therapeutic strategy has been proposed in many cancers. This approach was able to decrease GB growth in both in vitro and in vivo models [69]. The mesenchymal GB subtype [70] and those with aberrant c-MYC signaling [71] are particularly sensitive to GLS inhibition. Moreover, GLS plays a role in resistance to mTOR inhibitors, and dual inhibition of mTOR and GLS in vivo synergistically slowed GB growth [72]. A phase II clinical trial evaluating telaglenastat, a GLS inhibitor, in combination with poly(ADP-ribose) polymerases (PARPs) inhibitor, talazoparib, has been completed in advanced/metastatic solid tumors patients (NCT03875313).

Another potential therapeutic target is glutamate dehydrogenase (GDH). In GB, GDH expression is upregulated and correlates with poor patient prognosis. Inhibition of GDH decreased tumor proliferation in various in vitro and in vivo models [73]. Recently, researchers distinguished the two existing GDH isoforms (GDH1 and GDH2) and showed a correlation between strong GDH2 expression and improved patient prognosis. In vitro, overexpression of GDH2 leads to a significant decrease in proliferation, migration and clonogenicity of GB cells. It seems that it is GDH1, but not GDH2, that should be targeted in anticancer strategies [74].

Overall, targeting glutamine metabolism presents a potential avenue for developing effective therapies for GB and other cancers [68].

#### 2.2.2. Arginine Metabolism

Arginine is an amino acids actively metabolized by tumor cells to facilitate tumor progression and immunosuppression. L-arginine is a crucial component of the urea cycle and plays a role in modulating both immune function and tumor metabolism. Arginine is catalyzed by two enzymes: Arginase 1 (ARG1), which converts L-arginine into urea and ornithine; and cytokine-inducible Nitric Oxide synthase (NOS) which converts L-arginine into citrulline and Nitric Oxide (NO). In GB, arginine metabolism is altered, characterized by an upregulation of amino acid transporters CAT-1 and the Arginase enzymes, particularly ARG II located in the mitochondria. Additionally, there is a downregulation of key enzymes responsible for endogenous arginine synthesis, such as Argininosuccinate Synthase 1 (ASS1). Potential therapeutic strategies targeting arginine metabolism include targeting the Arginase enzyme and arginine depletion therapy (Figure 4).

Depleting extracellular arginine has been established as an effective treatment approach for cancers exhibiting deficiencies in arginine metabolism and dependence on exogenous sources. However, the arginine deficiency in the tumor microenvironment inhibits T-cell function, promoting immunosuppression. Thus, targeting arginine metabolism may present a dilemma between deprivation and replenishment.

Some GB tumors exhibit a deficiency in *ASS1*. These tumors account for 20% of all GB cases and are particularly sensitive to arginine deprivation. Depleting arginine using pegylated arginine deiminase (ADI-PEG20) has been shown to reduce tumor growth in vitro and in vivo [75]. The phase I study on recurrent high-grade gliomas showed promising results with ADI-PEG20 in combination with chemotherapy [76,77]. Targeting the Arginase enzyme has also been evaluated in clinical trials. INCB001158, an Arginase inhibitor, has completed a phase I/II clinical trial in combination with chemotherapy for the treatment of advanced solid tumors (NCT03314935). Further research is needed to determine the most effective strategy to target arginine metabolism in GB.

### 2.3. Lipid Metabolism

GB cells are surrounded by a lipid-rich environment, and lipid metabolism plays a critical role in the development and progression of GB. Lipids comprise a heterogeneous group of organic compounds, including fatty acids, triglycerides, phospholipids and cholesterol. Tumor reprogramming of lipid metabolism involves alterations in import/export pathways, lipid catabolism including Fatty Acid Oxidation (FAO), as well as alterations in de novo lipid synthesis pathways such as lipogenesis and cholesterol synthesis. Numerous potential therapeutic strategies targeting lipid metabolism have been investigated for the treatment of GB (Figure 5) [78].

#### 2.3.1. Lipid Droplets

In GB, lipid biosynthetic activity is accompanied by an accumulation of lipid droplets, which are particularly overexpressed in GSCs. These droplets contain large quantities of neutral lipids, triglycerides, and/or cholesterol esters, representing an important energy reserve that can be used in response to metabolic stress [79]. Furthermore, as these droplets are not detectable in healthy tissue, they have the potential to serve as a diagnostic biomarker for GB [80]. Interestingly, researchers have discovered that glucose deprivation can lead to the binding of choline kinase (CHK) alpha 2 to lipid droplets, which promotes lipid droplet lipolysis, increased FAO, and brain tumor growth [81]. Therefore, CHK alpha 2 is a promising therapeutic target for GB.

#### 2.3.2. De Novo Lipogenesis

Lipid biosynthetic activity is required to promote membrane biogenesis and to produce lipid signaling molecules that support the important proliferative activity of cancer cells [82]. Two main sources are used for de novo fatty acid synthesis: acetyl-CoA and acetate. Acetyl-CoA can be synthesized either from citrate through ATP-citrate lyase (ACLY) or acetate through acetyl-CoA synthetase 2 (ACSS2). This requires NADPH-reducing equivalents, which may be generated by the enzymatic activity of the malic enzyme or isocitrate dehydrogenase 1 (IDH1) or through the PPP or the serine and glycine metabolism pathways. De novo lipid synthesis is followed by an ATP-dependent acetyl-CoA carboxylation by acetyl-coA carboxylase (ACC), a limiting step, generating malonyl-CoA. Fatty acid synthase (FASN) then catalyzes the synthesis of palmitate from seven molecules of malonyl-CoA and one molecule of acetyl-CoA while consuming 7 molecules of ATP and 14 of NADPH. Acetate is a major fuel for the TCA cycle through the activity of the enzyme acetyl-CoA synthetase 2 (ACSS2) which generates acetyl-CoA. In GB, higher expression levels of ACSS2 are associated with higher tumor grade and lower survival rates. In mice, injection of *ACSS2* shRNA in GB cells significantly inhibits tumor growth [15]. ACSS2 expression appears to be critical for promoting GB viability and tumor growth by increasing the processes of mitochondrial TCA cycle and lipogenesis. In vitro, ACSS2 knockdown has led to the inhibition of self-renewal and the induction of cell death in GSCs [15]. In GB cell models, forced expression of EGFRvIII resulted in increased de novo lipogenesis and was correlated with increased tumor growth. This effect was reversed by inhibiting the ACC enzyme using siRNA [83]. Another potential GB therapeutic target includes the enzyme FASN. FASN expression was correlated with tumor grade and its inhibition resulted in anticancer effects in different in vitro models. Cerulenin and orlistat, drugs used in the treatment of obesity, inhibit FASN and showed anticancer activity in different GB cell models [84,85]. Further study on GSCs showed that cerulenin-mediated inhibition of FASN decreased proliferation, migration and expression of stem cell markers and increased the expression of GFAP, a marker of differentiation [86]. Thus, it appears that fatty acid synthesis plays a major role in the maintenance of GSCs. Currently, FASN inhibitors have already completed clinical trials in many cancers, including GB in combination with bevacizumab (NCT03032484, phase II). Rich JN and colleagues discovered that GSCs rely on polyunsaturated fatty-acid synthesis to maintain the integrity of their cellular membranes. By targeting the key enzyme elongation of very long chain fatty acids 2 (ELOVL2), involved in this process, researchers disrupted EGFR signaling and slowed down GSC proliferation [87]. Another targets for GSCs are stearoyl-CoA desaturase (SCD) or fatty acid desaturase 2 (FADS2) [88,89]. Targeting these pathways was found to increase palmitate accumulation and enhance TMZ sensitivity. YTX-7739, an SCD inhibitor, has been shown to be effective in treating GSCs either alone or in combination with TMZ by triggering lipotoxicity and impairing DNA damage repair. Interestingly, the study also revealed that aberrant MEK/ERK signaling and AMP-activated protein kinase (AMPK) repression are involved in SCD inhibitor sensitivity, while activation of AMPK determines treatment resistance [88].

#### 2.3.3. Fatty Acid Oxidation

Inhibition of FAO has also emerged as a promising therapeutic strategy for cancer treatment. FAO fuels cancer growth by generating ATP and NADPH. In GB, overexpression of fatty acid transporters has been observed such as the carnitine palmitoyltransferase (CPT) transporters, CPT1A and CPT1C, playing a key role in the transport of long-chain fatty acids from the cytoplasm into the mitochondria. Fatty acid binding protein 7 (FABP7) have also been identified as a potential marker for GSCs [90]. In different models of GB, both in vitro and in vivo, CPT1 inhibition using etomoxir [91], or FABP7 inhibition using SB-FI-26 or PPAR antagonists, have showed anticancer activity [79,92]. Recent research has linked high levels of mitochondrial FAO enzymes (CPT1A, CPT2, and ACAD9, the acyl-CoA dehydrogenase family member 9) to poor prognosis in recurrent GB patients. The study also revealed that inhibiting FAO metabolism reduces the expression of the immune checkpoint protein CD47, which is highly expressed in GSCs, and hinders tumor growth. Preclinical experiments have demonstrated that the combination of etomoxir, an inhibitor of FAO metabolism, and anti-CD47 antibodies effectively decreased GB growth [93].

#### 2.3.4. Cholesterol Metabolism

Cancer cells have a higher dependency on cholesterol compared to healthy cells for their survival and proliferation. Cholesterol can be obtained from the microenvironment or synthesized de novo by cancer cells. In healthy tissue, astrocytes rely on de novo synthesis, whereas GB cells inhibit their own synthesis and instead exhibit a higher dependence on exogenous cholesterol. This dependence is supported by an upregulation of low density lipoprotein receptors (LDLRs), which facilitate cholesterol uptake by GB cells. Both synthetic liver X receptors (LXR) agonists and statins, which block the activity of HMG-CoA reductase, have been suggested as potential anticancer agents due to their ability to modulate cholesterol metabolism [94].

GB cells have shown to be sensitive to LXR agonist-dependent cell death. Oxysterols are derivatives of cholesterol that serve as endogenous ligands for LXRs. The activation of LXRs promotes cholesterol efflux via sterol transporters, and inhibits cholesterol uptake by promoting LDLR degradation. This negative feedback system plays an essential role in maintaining cholesterol homeostasis. In a study, the authors showed that LXR-623, a LXRα-partial/LXRβ-full agonist selectively induced cell death in GB cells, leading to tumor regression and increased survival in animal model. This approach specifically targets the dysregulated cholesterol homeostasis in cancer cells, without affecting healthy cells [95].

A study showed that GSC populations exhibit overexpression of HMG-CoA reductase (HMGCR) and other components of the mevalonate pathway, which are required for de novo cholesterol synthesis [96]. Statins can inhibit HMGCR activity and thus de novo cholesterol synthesis. A phase II clinical trial (NCT02029573) investigated the efficacy of atorvastatin in combination with standard treatment in newly diagnosed GB patients. Although atorvastatin was well tolerated, it did not result in a significant clinical improvement. Interestingly, the trial identified high LDL levels as an important independent prognostic factor of poor cancer-related outcomes in the study cohort [97].

In addition to its role in de novo cholesterol synthesis, the mevalonate pathway is also involved in the prenylation of proteins. Farnesyl pyrophosphate (FPP) and geranylgeranyl pyrophosphate (GGPP) are precursors of cholesterol in this pathway and are implicated in the prenylation and activation of oncogenic proteins such as Ras and Rho. These proteins are crucial for the development and progression of cancers. Therefore, inhibiting the mevalonate pathway using farnesyl transferase inhibitors (FTIs) or geranylgeranyl transferase inhibitors (GGTIs) is a potential therapeutic approach for treating GB and other cancers [98,99,100,101].

In conclusion, in various in vitro and in vivo models, blocking cholesterol absorption or de novo biosynthesis has demonstrated anticancer effects [94].

#### 2.3.5. Other Lipid-Related Pathways

Methionine depletion was found to decrease the proliferation and increase cell death of GSCs. The depletion led to global DNA demethylation, reduced expression of stem cell markers, and decreased cholesterol synthesis through an alteration of the SREBF2-FOXM1 and ACA43 axis [102].

Some lipid-related proteins are highly expressed in GB but not detected in healthy glial cells. One such protein is very long-chain acyl-CoA synthetase homolog 3 (ACSVL3), which adds coenzyme A to fatty acids required for their β-oxidation. ACSVL3 expression is particularly upregulated in GSCs through the EGFR, HGF/c-MET, and AKT signaling pathways. Targeting ACSVL3, which is upregulated in GSCs, may be a promising therapeutic approach for GB [103,104].

Another study has identified sphingomyelin phosphodiesterase 1 (SMPD1), an enzyme that regulates the conversion of sphingomyelin to ceramide, as a drug target in GB. Fluoxetine has been found to inhibit SMPD1 activity and kill GB cells, inhibiting EGFR signaling and activating lysosomal stress. Combining fluoxetine with TMZ has led to complete tumor regression in mice [105].

Oncogenic pathways promote the expression of Sterol Regulatory Element-Binding Proteins (SREBPs), a family of transcription factors that regulate cholesterol and fatty acid metabolism, including lipogenesis. The SREBP-1a protein can activate different target genes involved in lipid synthesis while SREBP-1c mainly regulates fatty acid metabolism, for example by regulating FASN. The SREBP-2 protein mainly regulates the transcription of genes related to cholesterol metabolism, such as the genes encoding HMGCR and LDLR. The EGFR-PI3K-AKT pathway is a signaling pathway that regulates lipid biosynthesis. Indeed, it increases glucose uptake in cancer cells, thus promoting N-glycosylation of the SCAP protein, which is involved in the proteolytic cleavage activation of the SREBP-1 protein. In vivo, lapatinib-mediated inhibition of EGFR-PI3K-AKT signaling [106] or inhibition of the SCAP N-glycolysation [107] decreased tumor growth in EGFRvIII overexpressing GB models. Other molecules that impair the synthesis, maturation or activity of SREBP (e.g., betulin, quercetin or oxysterols) showed anticancer activity on GB cells [108]. A recent study investigated the effects of TAK901, an Aurora kinase inhibitor, on GB both in vitro and in vivo. The results showed that TAK901 reduced self-renewal and migration capacity of GSCs and induced cell cycle arrest and apoptosis by altering SREBP1-mediated lipid metabolism [109]. Impairment of cholesterol esterification through inhibition of sterol O-acyltransferase 1 (SOAT1) has also led to inhibition of SREBP-1 regulated fatty acid synthesis, impacting tumor growth in GB [110].

A study showed that GPD1, a member of the NAD-dependent glycerol-3-phosphate dehydrogenase family, is expressed specifically in dormant GSCs responsible for tumor relapse after chemotherapy. GPD1 plays a critical role in carbohydrate and lipid metabolism by catalyzing the reversible conversion of Dihydroxyacetone Phosphate and NADH to glycerol-3-phosphate and NAD^+^. Dormant GSCs upregulate the glycerophospholipid metabolism pathway that is dependent on GPD1, as well as the taurine and hypotaurine pathway, which both contribute to lipid metabolism and help to maintain GSC dormancy and stress resistance. Inhibiting GPD1 resulted in impaired GSCs maintenance and prolonged animal survival, making it a promising therapeutic target for treating GB [111].

Recent studies have shown that CD36 is selectively used by GSCs to promote their maintenance. Furthermore, CD36 expression is negatively correlated with patient prognosis and is an informative biomarker for malignancy. CD36 is a scavenger receptor that plays a crucial role in sensing danger-associated molecular patterns and oxidized lipoproteins. Therefore, CD36 has the potential to serve as both a biomarker for clinical diagnosis/prognosis and a promising target for cancer therapy [112].

AMPK is a ubiquitous enzyme which serves as a metabolic checkpoint, linking growth factor signaling to cell metabolism, partly through the negative regulation of mTOR. In EGFRvIII-expressing GB, the 5-Aminoimidazole-4-carboxamideribonucleotide (AICAR), an AMPK agonist, reduced tumor growth mainly through the inhibition of cholesterol and fatty acid synthesis [113].

Overall, lipid metabolism, including de novo lipogenesis, FAO, cholesterol metabolism and others lipid-related pathways, plays a crucial role in the growth and survival of GB cells, and targeting these pathways may represent potential therapeutic options [79].

### 2.4. Nucleotide Metabolism

Targeting nucleotide metabolism is a potential therapeutic strategy in the treatment of GB (Figure 6). Nucleotides and deoxynucleotides are essential building blocks of DNA, RNA, and ribosomes. There are also involved in energy metabolism (ATP, GTP) and main coenzymes (NAD^+^, NADP^+^, FAD, CoA), making them crucial for cellular signaling. These molecules are generated through two primary pathways: de novo synthesis and nucleotide salvage pathways. De novo synthesis of purines or pyrimidines requires ribose, amino acids, and significant amounts of energy, while nucleotide salvage pathways require less energy and use pre-formed purines or pyrimidines from nucleotide catabolism.

The salvage pathway is the primary source of pyrimidines for differentiated or quiescent cells, while proliferating cells are expected to rely more on de novo synthesis to meet their increased pyrimidine requirements. One study showed that a subpopulation of GSCs is sensitive to the inhibition of de novo pyrimidine synthesis [114], highlighting its potential therapeutic targeting in GB treatment. Enzymes necessary for the de novo biosynthesis of pyrimidines, such as dihydroorotate dehydrogenase (DHODH) and uridine monophosphate synthase (UMPS), are overexpressed in GB cell lines and patient tumors. DHODH inhibition impaired GB cell proliferation, including those resistant to TMZ. Interestingly, in vivo, inhibition of DHODH did not affect pyrimidine levels in healthy brain cells, emphasizing the activity of the pyrimidine salvage pathway in these cells [115,116]. A recent study found that BAY2402234, a drug that inhibits the DHODH pathway, successfully suppressed the growth of GSCs in vitro and reduced GB growth in human GB xenograft models [117]. Another study also found that GSCs depend on de novo pyrimidine synthesis and linked a metabolic aberration to driver mutations. Indeed, EGFR or PTEN mutations activated carbon influx through pyrimidine synthesis. Targeting enzymes involved in pyrimidine synthesis, such as CAD protein (carbamoyl-phosphate synthetase 2, aspartate transcarbamylase, and dihydroorotase) and DHODH, inhibited GSC survival and self-renewal and reduced tumor growth in rodent models. Inhibition of both tumor-specific driver mutations and DHODH activity demonstrated sustained inhibition of pyrimidine synthesis and tumorigenic capacity. The study suggests a potential therapeutic approach for precision medicine by targeting metabolic reprogramming and driver mutations concomitantly to completely inhibit specific pathway [118]. The activity of the pyrimidine salvage pathway in GB, highlighting by increased localization of radiolabeled tracer 18F-fluorothymidine (FLT) within the tumor, suggests potential prognostic value. However, further research is needed to fully understand the mechanisms underlying pyrimidine salvage pathway activity in GB, and to develop effective strategies for targeting this pathway [119].

Both the de novo and salvage pathways for purine synthesis have been identified in GB. De novo purines synthesis depends on c-MYC and plays an essential role in cell proliferation and the maintenance of GSCs tumorigenic properties [120]. A study demonstrated that intracellular levels of purines, particularly guanosine, are linked to radiation resistance due to the resulting GTP synthesis. In GB, the rate-limiting enzyme for de novo guanosine synthesis, inosine monophosphate dehydrogenase (IMPDH2), is upregulated, resulting in increased GTP biosynthesis, which is associated with a poor prognosis. Mycophenolic acid (MPA) and its prodrug mycophenolate mofetil (MMF) are FDA-approved inhibitor of GTP synthesis. Inhibiting GTP synthesis can sensitize GB cells to radiation by disrupting DNA repair. On the other hand, purines administration in radiosensitive models has been shown to lead to acquired resistance to radiotherapy [121]. Further research has found that the ciliary protein ARL13B interacts with IMPDH2 and contributes to chemoresistance by inhibiting the purine salvage pathway. It has been demonstrated that MMF can inhibit the ARL13B-IMPDH2 interaction, which enhances the effectiveness of TMZ [122]. Purine salvage pathway activity has also been observed in GB, including the hypoxanthine salvage pathway, which has been correlated with anti-folate resistance [121].

The potential of inhibiting nucleotide metabolism as a therapeutic strategy for GB treatment is currently being evaluated. One approach involves using gemcitabine, a cytidine analogue that has shown promising results in the treatment of solid tumors, including high-grade gliomas. Gemcitabine integrates into the DNA of proliferating cells and inhibits ribonucleotide reductase, thus hindering DNA synthesis. Since gemcitabine can pass through the blood–brain barrier, it can accumulate in brain tumors. A phase II clinical trial combining gemcitabine with radiotherapy in high-grade glioma patients demonstrated its efficacy [123]. However, the short half-life, side effects, and chemoresistance of gemcitabine have been reported [124].

In conclusion, the inhibition of nucleotide metabolism and the use of nucleotide-based therapies represent a promising avenue for the development of new GB treatments.

### 2.5. Monocarbon Metabolism

In GB, monocarbon units are essential for nucleotide synthesis, methylation reactions, and maintenance of redox potential, supporting tumor growth [125]. Metabolic vulnerabilities have been identified in certain conditions involving one-carbon metabolism. In poorly vascularized tumor regions, cancer cells depend on serine and glycine metabolism for survival, highlighted by high levels of mitochondrial serine hydroxymethyltransferase (SHMT2) and glycine decarboxylase (GLDC). SHMT2 inhibits PKM2 activity, reducing oxygen consumption and creating a metabolic state that promotes the use of upstream metabolic intermediates. GLDC inhibition impaired cancer cells survival with high SHMT2 levels due to an excess of glycine, not metabolized by GLDC, which is thus converted to toxic molecules (aminoacetone and methylglyoxal). Therefore, SHMT2 plays a crucial role in the adaptation of cancer cells to tumor microenvironment, while also rendering them vulnerable to inhibition of the glycine cleavage system [126].

Another metabolic vulnerability involves 5-methylthioadenosine phosphorylase (MTAP). Homologous deletions of *MTAP* are found in 40% of GB cases, resulting in a dependence of cancer cells on Protein Arginine Methyltransferase 5 (PRMT5). In these *MTAP*-deficient cancer cells, inhibition of PRMT5 inhibits tumor growth [127]. Other studies have demonstrated that PRMT5 inhibition effectively suppressed tumor growth in both differentiated GB cells and GSCs [125] and prolonged the survival of patient-derived xenograft models [128]. A phase I clinical trial is ongoing to evaluate the value of a PRMT5 inhibitor in treating solid tumors, including GB (NCT02783300).

One possible approach to target monocarbon metabolism is through dietary intervention. Studies have shown that a low methionine diet can reduce circulating antioxidant and nucleotide levels, and enhance the sensitivity of tumors to radio- and chemo-therapy. Therefore, implementing a low methionine diet may represent a promising adjunctive therapy for GB treatment [129].

Folate is an important source of one-carbon units. Anti-folate drugs, such as methotrexate (MTX) and pemetrexed, have shown promising results in GB and are being considered as a therapeutic option in combination with other treatments. They have been found to be selectively toxic to GSCs but not to normal fibroblasts or neural stem cells. In an in vivo experiment, MTX alone failed to show anti-GSC effects but enhanced the effects of CEP1347, an inducer of GSC differentiation. Combining anti-folate drugs with cytotoxic and differentiation therapies could lead to a new and effective way to eliminate GSCs, offering a promising treatment for GB patients [130].

### 2.6. Nicotinamide Metabolism

Pharmacological inhibition of nicotinamide phosphoribosyltransferase (NAMPT), an enzyme essential for NAD^+^ biosynthesis has shown promising results in cancer therapy by repressing glycolytic phenotype [131]. Further, a study showed increased sensitivity of MYC-amplified GB models to glycolysis inhibition mediated by NAMPT inhibitor, highlighting a metabolic vulnerability in these tumor subtypes [132].

The modulation of DNA methylation is crucial for cancer cells and nicotinamide metabolism plays a fundamental role in this process. Nicotinamide N-methyltransferase (NNMT) was among the most consistently overexpressed metabolic genes in GB compared to healthy brain tissue and is implicated in methionine metabolism. NNMT was preferentially expressed in mesenchymal GSCs. The depletion of S-adenosyl methionine (SAM), a methyl donor generated from methionine, by NNMT leads to lower levels of methionine, SAM, and nicotinamide, but higher levels of oxidized NAD^+^ in GSCs compared to differentiated tumor cells. Targeting NNMT expression reduced the proliferation and self-renewal of mesenchymal GSCs, and reduced tumor growth in vivo. The results showed also that NNMT could be a potential therapeutic target for GB by disrupting the oncogenic DNA hypomethylation [133].

### 2.7. TCA Cycle

Inhibiting the TCA cycle, which is essential for both catabolic and anabolic functions necessary for tumor growth, is a promising strategy for effective cancer therapy.

CPI-613 is a first-in-class drug that targets the TCA cycle. Specifically, it is a lipoate analogue that inhibits two major TCA cycle enzyme complexes, α-ketoglutarate dehydrogenase (α-KGDH) and pyruvate dehydrogenase (PDH), leading to accelerated and inefficient consumption of nutrient stores in cancer cells [134]. Although the exact mechanism by which CPI-613 exerts its anticancer effects is not fully understood, it has demonstrated promising results in phase I and II clinical trials for various types of cancer (NCT01832857) and was evaluating in phase III clinical trials for other malignancies (NCT03504410, NCT03504423). These findings suggest that targeting GB metabolism could be achieved through a novel class of TCA-targeted therapy, such as CPI-613. However, further research is needed to confirm its effectiveness in GB [135].

IDH catalyzes the oxidative decarboxylation of isocitrate to α-ketoglutarate and is involved in multiple cellular functions, such as glucose sensing, lipogenesis, glutamine catabolism, and defense against ROS and radiation [136,137]. Mutations in *IDH* occur early in tumorigenesis and are specific to tumors, providing an attractive therapeutic target in gliomas. The new WHO classification excludes the diagnosis of GB in the presence of *IDH* mutations. Nevertheless, in vitro studies have shown that targeting IDH1 wild-type could be a potential strategy to sensitize cells to radiation therapy. Knockdown of wild-type *IDH1* in GB cells resulted in decreased levels of NADPH, deoxynucleotides, and antioxidants, demonstrating the potential of IDH1 targeting as a therapeutic approach for GB [138].

### 2.8. Electron Transport Chain and Oxidative Phosphorylation

Analysis of patient high-grade gliomas revealed that 43% of tumors contained at least one mitochondrial DNA alterations in genes encoding complexes I, III and/or IV of the respiratory chain [139]. Strong expression of cytochrome c oxidase (CcO), a terminal enzyme in the respiratory chain, has been correlated with poor patient outcomes [140]. Further research on CcO has demonstrated that COX4-1 (an isoform of CcO subunit 4) is involved in cell proliferation, repression of ROS production, increased expression of stem cell markers, and self-renewal of GSCs [141]. Additionally, inhibition of the respiratory chain or depletion of mitochondrial DNA in cancer cells increases CD133 expression, a stem cell marker [142].

Studies have shown that OXPHOS inhibition leads to the downregulation of insulin-like growth factor 2 mRNA-binding protein 2 (IMP2), resulting in the inhibition of GSC maintenance. A strong correlation was observed between IMP2 expression, self-renewal of CGC, and decreased survival in animal models [143]. Induction of MPC1 expression may be a potential therapeutic strategy for GB, as it has been linked to increased chemosensitivity to TMZ [144].

Metformin, an anti-diabetic drug, has shown promise as a therapeutic option for GB, with multiple anticancer effects including the inhibition of complex I of the respiratory chain, activation of AMPK, suppression of the mTORC1 pathway, alteration of mitochondrial biosynthetic pathways, and stimulation of the immune system [145,146,147]. Metformin has been shown to reduce cell viability, proliferation, and migration, increase apoptosis, disrupt epithelial-mesenchymal transition, increase the production of ROS and negatively impact mitochondrial membrane potential and biogenesis in certain GB cell lines, including GSCs [148,149]. Notably, metformin has demonstrated higher antiproliferative activity on CD133-expressing subpopulations, suggesting a certain level of selectivity towards GSCs [150]. Clinical trials evaluating metformin in GB have been conducted, including a completed study on low dose TMZ plus metformin or placebo in patients with recurrent or refractory GB (NCT03243851). Phase II clinical trials are currently in progress to assess the effectiveness of metformin in combination with chemotherapy, radiotherapy, ketogenic diet, and/or paxalisib (PI3K/mTOR inhibitor) in GB patients [151].

### 2.9. Transporters and Ion Channels

In GB, a decrease in the expression of the mitochondrial pyruvate transporters MPC1 and MPC2 has been observed. This leads to a decrease in the flow of pyruvate into the mitochondria and an increase in glycolytic activity and compensatory pathways that maintain fuel oxidation in the TCA cycle, such as glutaminolysis, FAO, and branched chain amino acid pathways [152]. A decrease in MPC1 expression has been associated with a poor response to TMZ in GB [144]. Thus, inducing the expression of MPC1 could represent a potential therapeutic strategy for GB.

The voltage-dependent anion channel 1 (VDAC1), a mitochondrial protein controlling cell energy, metabolic homeostasis and apoptosis, is one of the relevant targets in GB. Depleting VDAC1 expression using short interfering RNA inhibit GB growth and tumor growth in xenograft mouse models. VDAC1 depletion also reversed oncogenic properties and altered transcription factors, leading to tumor cell differentiation into neuron- and astrocyte-like cells [153].

## 3. Complexity, Heterogeneity, and Plasticity of Tumor Metabolism

Tumor metabolism is a complex and dynamic process, characterized by heterogeneity and plasticity. The metabolic heterogeneity of GB is due to a combination of intrinsic factors such as tissue of origin and genetics, as well as extrinsic factors such as the patient’s metabolism and the microenvironment [13]. Different microenvironments and cell subpopulations in GB lead to metabolic diversity, and cancer cells evolve over time in response to a constantly changing environment, highlighting the notion of metabolic complexity and flexibility.

### 3.1. Metabolic Heterogeneity and Plasticity

This heterogeneity has been demonstrated through a multitude of studies in xenograft models or from resected patient tumors using ^13^C or multi-omics approaches [154]. Garofano et al. employed a multi-omics computational approach to analyze single GB cells and patient tumors and identified four subtypes of GB based on neurodevelopmental (proliferative/progenitor vs. neuronal) and metabolic (mitochondrial vs. glycolytic/plurimetabolic) criteria. This study revealed that different subpopulations of GB cells exhibit distinct metabolic profiles, with some being dependent on OXPHOS and others relying on aerobic glycolysis, amino acid metabolism, and lipid metabolism and showed that mitochondrial GB was associated with a more favorable clinical outcome [155]. Another multi-omics study defined a new classification of GB based on the tumor immune landscape, which effectively predicts patient outcome and defines specific lipid metabolism for each subtype [156]. Recent research has highlighted the metabolic heterogeneity of GB, with GSCs and non-GSCs exhibiting distinct metabolic profiles. GSCs are characterized by lower lipid droplet accumulation and a distinct lipid metabolism, with decreased levels of neutral lipids and increased polyunsaturated fatty acid production [157]. Other studies showed that cancer cells in nutrient-deprived regions metabolize droplet lipids more extensively [158], while cancer cells in oxygen-deprived regions exhibit a more glycolytic profile [55]. Further research has revealed metabolic heterogeneity even within the GSC population. One study showed that mesenchymal and proneural GSCs have different metabolic profiles and responses to metabolic-related therapies. The study found that mesenchymal GSCs were more glycolytic and less responsive to metformin, while proneural GSCs were less invasive, metabolized glucose through the Pentose Phosphate Pathway, and were more responsive to metformin. These results suggest the importance of considering metabolic heterogeneity in future clinical trials, as targeting glycolysis may be an effective strategy for inhibiting mesenchymal tumor cells, while proneural cells may respond better to OXPHOS inhibition [159]. Metabolic plasticity enables cells to adapt to various environmental factors, such as fluctuations in nutrient and oxygen availability, oxidative stress, and therapeutic interventions. One example of this plasticity was highlighted in an in vitro study which demonstrated that the inhibition of coenzyme Q biosynthesis with 4-nitrobenzoate (4-NB) significantly increased the cholesterol content in glioma cells, leading to decreased oxygen levels, reduced plasma membrane fluidity, and stabilization of HIF-1α, thereby increasing glycolysis [160]. This study revealed metabolic plasticity and the interconnection between OXPHOS metabolism, glycolysis, and cholesterol metabolism. The ability of tumor cells to adapt their metabolism highlights the importance of understanding metabolic changes in cancer in order to develop therapies that impair tumor cell adaptation.

### 3.2. Metabolic Interactions in the Tumor Microenvironment

In addition to metabolic heterogeneity within tumors, cancer cells have been found to engage in cooperative metabolic interactions to meet their metabolic requirements. This cooperation can occur between different cancer cells as well as between cancer cells and stromal cells. For example, glycolytic and oxidative cancer cells, even located far from each other, have shown to mutualize their energy resources [55]. There is also evidence of cooperation between cancer cells and endothelial cells to promote neoangiogenesis, ensuring the availability of essential nutrients and oxygen to cancer cells [161]. The immune cells have also demonstrated their ability to regulate tumor metabolism and cooperate with cancer cells. M2 macrophages have shown to secrete interleukin-6 (IL-6), promoting the phosphorylation of Phosphoglycerate Kinase 1 (PGK1) in tumor cells mediated by 3-phosphoinositide-dependent protein kinase 1 (PDPK1). This phosphorylation facilitates glycolysis, leading to increased tumor cell proliferation and tumorigenesis [162]. A metabolic symbiotic relationship between cancer cells and myeloid-derived suppressor cells (MDSCs) has also been described, where lactate produced by cancer cells can sustain MDSCs, which play a role in creating an immunosuppressive microenvironment that impacts both innate and adaptive immunity [163]. In conclusion, the metabolic interactions between cancer cells and surrounding cells play a crucial role in regulating tumor metabolism and can have protumorigenic effects. Further research should explore these interactions in order to develop innovative cancer treatment strategies.

## 4. Combination of Therapies

### 4.1. Metabolic-Related Therapy

Metabolic-related therapy has been considered as a promising approach for cancer treatment due to the metabolic alterations present in cancer cells. However, its efficacy as a single therapy has been limited. Combining metabolic therapy with other treatments has shown greater potential. Despite the potential benefits, the metabolic plasticity and heterogeneity of cancer cells can result in therapeutic resistance and undermine the effectiveness of metabolic therapy. To overcome these challenges, it is essential to comprehend the underlying mechanisms of metabolic heterogeneity and plasticity in cancer cells and to determine the metabolic dependencies and preferences of different cancer subtypes, as well as the compensatory metabolic mechanisms.

A study conducted by Yang and colleagues shed light on the metabolic plasticity of GB cells in response to downregulation of mitochondrial pyruvate transporters carriers MPC1 and MPC2. This resulted in a decrease in the transport of pyruvate into the mitochondria, thereby activating alternative sources of fuel, such as glutaminolysis, branched-chain amino acid pathways, and FAO, for the TCA cycle [152]. The concurrent inhibition of MPC and GDH led to a significant inhibition of tumor growth compared to the use of either inhibition separately [140]. This underscores the importance of taking into account the adaptability of GB cells when designing therapeutic strategies.

Due to the metabolic plasticity of cancer cells, dual inhibition of tumor bioenergetics has been proposed as a relevant therapeutic approach. The dual inhibition of glycolysis and OXPHOS was investigated on GB tumorspheres. The combination of 2-DG and metformin resulted in a decrease in the invasive properties of cancer cells, prolonged survival in a mouse model, and led to a downregulation of stemness- and epithelial mesenchymal transition-related genes [164]. Targeting ATP synthesis through triple therapy with targeted inhibitors of the TCA cycle, phospholipids, and glycolysis (EPIC-0412, AACOCF3, and 2-DG) has also been found to inhibit tumor growth both in vitro and in vivo [165]. These results suggest that multiple inhibition of cellular bioenergetics may be a promising approach for GB treatment that requires further clinical evaluation.

The value of combining metabolic-targeted therapies was also demonstrated by Hoang-Minh et al., who found that *FABP7* knockout increased the sensitivity of GSCs to pharmacological inhibition of glycolysis induced by 2-DG [79]. The dual inhibition of glycolysis and FAO pathways, using DCA and ranolazine, has also shown inhibitory effects on tumor growth and increased survival in an orthotopic xenograft model [166]. The therapeutic potential of inhibiting both CPT1A and glucose-6-phosphate dehydrogenase (G6PD), critical enzymes for FAO and the PPP, respectively, was also evaluated. The combination of etomoxir and DHEA, inhibitors of CPT1A and G6PD, respectively, resulted in decreased viability, ATP levels, and expression of genes associated with stemness and invasiveness, and showed inhibitory effects on tumor growth and increased survival in a mouse model [167]. Another study revealed that the combination of cytoplasmic phospholipase A2 (*cPLA2*) knockdown and metformin impair mitochondrial energy metabolism in primary GB cells, and reduced tumor growth and prolonged survival in a patient-derived xenograft model [168]. A recent study found that the loss of branched-chain amino acid transaminase 1 (BCAT1) in GB leads to a metabolic vulnerability that can be targeted with α-ketoglutarate (AKG). The combination of BCAT1 inhibitor gabapentin and AKG was found to be synthetically lethal in patient-derived GB tumors both in vitro and in vivo. The loss of BCAT1 resulted in an imbalance of NAD^+^/NADH ratio and impaired OXPHOS, nucleotide biosynthesis, and mTORC1 activity. The combination of BCAT1 loss and AKG treatment led to mitochondrial dysfunction, cellular building block depletion, and cell death, providing a targetable metabolic vulnerability in GB and a potential therapeutic strategy to improve treatment outcomes [169].

### 4.2. Metabolic Therapies and Radiotherapy

Metabolic therapies and radiotherapy have been shown to be a promising combination in treating GB. Resistance to radiotherapy has been linked to high glycolytic states and increased mitochondrial reserve capacity [170].

Inhibition of glycolysis has been shown to radiosensitize GB cells in various in vitro and in vivo models, either through knockdown of *HK2* [171], use of 2-DG [30] or DCA [50]. Additionally, depletion of glutathione (via GLS inhibition), NAD^+^ (via ascorbate use and PARPs inhibition) or inhibition of IDH or nucleotide metabolism can also result in radiosensitization of GB cells [9].

The repair of radiation-induced DNA damage is a barrier to the efficacy of radiotherapy, and these DNA repair mechanisms are dependent on NAD^+^, a cofactor used by the key enzymes of the DNA repair pathways such as PARPs. Inhibiting PARPs is a therapeutic strategy being evaluated for the treatment of GB.

Other strategies to limit the availability of NAD^+^ have been tested, such as the inhibition of nicotinamide phosphoribosyltransferase (NAMPT). NAMPT is a rate-limiting enzyme for NAD^+^ recovery. NAMPT was clinically tested as a monotherapy but the trial was prematurely abandoned due to a narrow therapeutic index [172].

The IDH1 enzyme, which is overexpressed in GB, plays a role in the biosynthesis of NADPH, a critical reducing agent. Depletion of NADPH results in oxidative stress, which makes the cells more susceptible to radiation-induced DNA damage. By inhibiting wild-type IDH1, NADPH levels can be decreased, thus increasing the sensitivity of GB cells to radiation therapy, as demonstrated in various in vitro and in vivo studies [138].

Since the effects of radiotherapy on DNA are largely due to ROS (O_2_^−^, H_2_O_2_, OH^−^), a relevant therapeutic strategy is to increase the levels of ROS within cancer cells or to decrease the levels of antioxidants. In order to increase the levels of ROS, it is possible to act on the metabolism of glutamine, which is involved in the production of mitochondrial ROS through its oxidation in the TCA cycle. Glutamine metabolism plays a key role in the maintenance of redox homeostasis due to its role in the synthesis of glutathione, a major antioxidant. GLS inhibitors have been tested in IDH-mutated GB, which are known to be highly dependent on glutaminase for survival. CB-839, a GLS inhibitor, has been shown to sensitize cells to oxidative stress and radiotherapy in both in vitro and in vivo models [173]. CB-839 is currently being tested in combination with radiotherapy and TMZ in a phase I clinical trial for patients with IDH-mutated diffuse or anaplastic astrocytoma (NCT03528642). Ascorbic acid has been shown to increase ROS in cancer cells. The combination of TMZ, radiotherapy and ascorbic acid is currently being evaluated in a phase II clinical trial for the treatment of GB (NCT02344355) [174].

The combination of radiotherapy and the arginine-depleting agent ADI-PEG20 has been found to greatly improve treatment outcome of GB, resulting in a durable and complete response, with an extended disease-free survival in a GB model. ADI-PEG20 enhances the sensitivity of GB cells to radiation by generating cytotoxic peroxynitrites, which are nitrogen-based free radicals with potent cellular toxicity. Additionally, ADI-PEG20 promotes the infiltration of immune cells into the tumors and shifts the anti-inflammatory phenotype of these cells to a pro-inflammatory one. This leads to an increased in DNA damage and in a more aggressive immune response against the tumors [175].

Finally, the combination of radiotherapy and nucleotide metabolism inhibitors is another therapeutic strategy used in many cancers. Gemcitabine, a nucleoside analogue, has been tested in clinical trials in combination with radiotherapy for the treatment of high-grade gliomas. A phase II trial in patients with newly diagnosed GB showed a radiosensitizing effect of gemcitabine [123]. In some studies, gemcitabine has shown the ability to act synergistically with radiotherapy but also with other chemotherapeutic agents (e.g., paclitaxel, cisplatin, carboplatin). However, a short half-life, side effects and chemoresistance have been described [124].

In conclusion, the combination of metabolic therapies with radiotherapy has shown promising results. Inhibition of glycolysis, IDH or nucleotide metabolism, NAD^+^ depletion, use of ascorbic acid or ADI-PEG20 are some of the strategies that have been evaluated on GB.

### 4.3. Metabolic Therapies and Chemotherapy

TMZ induces remodeling of the respiratory chain [176] and has been shown to increase fatty acid uptake in GSCs [177]. In wild-type TP53 GB cells, TMZ induces an upregulation of TP53 which leads to a repression of PINK1 (PTEN-induced putative kinase 1) and subsequent mitophagy. An increase in mitochondrial mass after TMZ treatment has been observed, which is thought to be dependent on AMPK-mediated signaling. Inhibition of mitochondrial fusion process in GB cells sensitizes them to TMZ [178].

Inhibition of glycolysis has been demonstrated to enhance the efficacy of chemotherapy in high-grade glioma models. This effect has been observed both in vitro, through *HK2* knockdown [42], and in vivo, through local administration of 3-bromopyruvate (3-BP) [179]. A combination of the glycolysis inhibitor 3-bromo-2-oxopropionate-1-propyl ester (3-BrOP) and carmustine was shown to have a synergistic effect against GSCs under hypoxic conditions, which are highly resistant to standard chemotherapy agents such as TMZ or carmustine. This combination reduced the ability of GSCs to form neurospheres in vitro and to inhibit tumor growth in vivo [180]. The combination of 2-DG and l,3-bis(2-chloroethyl)-1-nitrosourea (BCNU) was found to increase the sensitivity of GB cells by regulating glycolysis, ROS, and Endoplasmic Reticulum stress pathways, resulting in increased energy deficiency, oxidative stress, and apoptosis [181]. A recent study revealed that the combination of TMZ and metformin effectively suppressed the proliferation and induced apoptosis of both glioma cells and GSCs. This treatment downregulated the AKT-mTOR signaling pathway while enhancing AMPK activation, reduced tumor growth in vitro and in vivo, and making it a promising therapeutic option for advanced GB [182]. Moreover, the combination of TMZ with energy metabolism inhibitors, gossypol and phenformin, resulted in significant impairment of energy production, viability, stemness, and invasiveness in GSC lines compared to TMZ monotherapy or gossypol-phenformin dual therapy [183].

The combination of the FAO inhibitor etomoxir with TMZ has also been studied. The study found that GB tissues had a higher expression of FAO-related genes compared to healthy brain tissue, and that the combination of etomoxir and TMZ had a more pronounced effect in reducing cell viability, stemness, and invasiveness, and in improving survival outcomes in mouse xenograft model [184]. A study showed that the accumulation of saturated fatty acids, particularly palmitate, combined with TMZ treatment improves its efficacy against GB cells. The inhibition of SCD and/or FADS2 enhances palmitate accumulation and increases TMZ efficacy. The combination therapy was effective against recurrent GB cells [89].

Moreover, a study has revealed that a long non-coding RNA (lncRNA), TP73-AS1, is overexpressed in primary GB patient samples and its expression in tumors is correlated with poor patient prognosis. TP73-AS1 promotes TMZ resistance through the regulation of metabolic genes such as *ALDH1*. The *ALDH1* gene, which encodes type 1 aldehyde dehydrogenases, is overexpressed in GSCs and has been implicated in tumorigenesis as well as in TMZ resistance [185]. Clinical trials combining an ALDH inhibitor (disulfiram) and copper with radio-chemotherapy for GB are underway (NCT02715609).

In conclusion, several metabolic approaches have been explored in combination with chemotherapy for the treatment of GB. These include targeting glycolysis, OXPHOS and fatty acid metabolism. This has shown promising results in preclinical studies and further investigation is warranted to determine their potential as effective treatments for GB.

### 4.4. Metabolic and Targeted Therapies

Targeted therapies, such as bevacizumab and PI3K-AKT-mTOR inhibitors have shown to have an impact on tumor metabolism. A study showed that treatment with bevacizumab led to a metabolic adaptation toward anaerobic metabolism, increasing glycolytic activity, lactate production, and decreasing TCA cycle metabolites in orthotopic GB models. The treatment also resulted in a decrease in glutathione levels, indicating oxidative stress within the tumors [186]. Currently, bevacizumab is the only approved targeted therapy for the management of relapsed GB; however, its combination with the standard protocol has not shown significant clinical benefits. Further research exploring the combination of bevacizumab with glycolytic inhibitors may be worthwhile [187].

The PI3K-AKT-mTOR pathway is a promising strategy for GB therapy as it plays a crucial role in regulating cell proliferation, metabolism, and survival. However, therapies targeting this pathway, either alone or in combination with radiochemotherapy, have demonstrated limited clinical efficacy. In this context, a study showed that inhibition of mTOR signaling can protect glioma cells from hypoxia-induced cell death in an autophagy-independent manner [188]. Therapy resistance to mTOR inhibition may be due to a metabolic adaptation in the tumor cells. A study showed that mTOR-targeted therapy led to an increase in the levels of GLS and glutamate, and that inhibiting GLS in conjunction with the treatment sensitized cells to mTOR inhibitors in various in vitro and in vivo models [72]. As previously mentioned, a phase II clinical trial evaluating telaglenastat, a GLS inhibitor, in combination with PARP inhibitor, talazoparib, has been completed in advanced/metastatic solid tumors patients (NCT03875313).

In conclusion, targeted therapies have been shown to have a significant impact on tumor metabolism. The combination of bevacizumab with glycolytic inhibitors may offer promising results, while the limited efficacy of therapies targeting the PI3K-AKT-mTOR pathway may be improved through glutaminase inhibition. Further research is necessary to fully explore these treatment options.

### 4.5. Metabolic Therapies and Immunotherapy

The tumor microenvironment of GB is highly immunosuppressive, limiting the effectiveness of immunotherapies. Lactate, produced through the Warburg effect in cancer cells, plays a significant role in establishing an immunosuppressive environment. Targeting lactate metabolism by inhibiting glycolysis or lactate export could improve the efficacy of immunotherapies for GB [189].

Combining immunotherapy with metabolic therapies has shown promise in treating GB. For example, the combination of T-cell therapy with a metabolism-modulating drug, liposomal avasimibe, has demonstrated improved antitumor efficacy in mouse models of GB [190]. Another example is the inhibition of the enzyme indoleamine 2,3 dioxygenase 1 (IDO1), which is frequently expressed in GB and leads to the inhibition of T lymphocytes’ cytotoxic functions. Inhibition of IDO has increased GB’s sensitivity to checkpoint inhibitors in various in vivo models [191]. However, early clinical trials incorporating the combination of immunotherapy and IDO inhibition were unsuccessful in the treatment of melanoma and pancreatic cancer. A recent study investigated the combination of anti-PD-1 immunotherapy with a novel glutamate modulator, BHV-4157, in a mouse model of GB. The results showed better survival when using BHV-4157 in combination with anti-PD1 immunotherapy. The concentration of glutamate in the tumor environment decreased, and there was an increase in CD4^+^ T cells, major players in the adaptive immune response, and a decrease in regulatory immune cells within the tumor. These findings suggest that glutamate in GB has a role in immunosuppression and provide a basis for further exploring combinatorial approaches for GB treatment [192].

### 4.6. Diet Interventions

Diet is a crucial environmental factor that modulates tumor metabolism, and emerging evidence suggests that dietary interventions could influence the therapeutic response of cancer patients. For instance, fasting for 48 h prior to radio- and chemotherapy has been shown to increase survival in GB xenograft models [193]. In addition, calorie-restricted ketogenic diets have been associated with a robust increase in CD8^+^ T cells, a key player in the antitumor adaptive immune response, as well as a decrease in the immune checkpoint ligand, PDL-1, highlighting the role of diet in immune defense against cancer [194]. Furthermore, low-methionine diets have been found to limit monocarbon metabolism, increase circulating antioxidant and nucleotide levels, and sensitize tumors to radio- and chemotherapies [129]. The calorie-restricted ketogenic diet and intermittent fasting remain the most extensively studied practices to date. A recent clinical trial, the RGO2 trial (NCT01754350), investigated the effects of these two dietary interventions on patients with recurrent GB undergoing radiotherapy. While the results indicated that a ketogenic diet or intermittent fasting practice leads to significant metabolic changes in patient serum samples, no significant differences were found in progression-free survival or overall survival. However, the caloric intake of the control group was not controlled and was lower than expected, leading to a potential bias in the results. Future research is needed to define the optimal protocols for fasting and restrictive diets, including the composition, intensity, and duration of dietary interventions, to ensure a favorable benefit-to-risk ratio for cancer patients [195]. Overall, dietary interventions have shown potential in modulating tumor metabolism and improving the therapeutic response of cancer patients, but further research is needed to determine the optimal protocols for their clinical application.

## 5. Discussion and Perspectives

Significant advancements in the field of cancer metabolism have provided new avenues for the treatment of GB, a highly aggressive brain tumor that remains challenging to treat effectively. Two key discoveries have driven interest in targeting tumor metabolism for anticancer strategies: the regulation of metabolic enzymes by oncogenes and tumor-suppressor genes, and the identification of specific mutations in genes encoding metabolic enzymes involved in tumorigenesis. Examples of such mutations include the succinate dehydrogenase mutation, linked to the development of paragangliomas [196], and the *IDH* mutations, commonly found in gliomas and serving as important diagnostic, prognostic, and predictive biomarkers for a favorable response to TMZ in glioma patients [10].

GB initiation and progression involve significant metabolic reprogramming to support rapid cell division and growth. Alterations in oncogenes and tumor-suppressor genes drive changes in the expression of key metabolic enzymes and transporters, leading to altered metabolism in cancer cells. Metabolic characteristics can directly impact signaling pathways and therapeutic responses [20]. The tumor microenvironment plays a pivotal role in shaping the metabolic profile of GB, as factors such as tumor growth, angiogenesis activation, hypoxic zone formation, and nutrient deprivation contribute to metabolic alterations. These changes enable cancer cells to adapt and meet their demands in response to the changing microenvironment. Moreover, GB cells may switch to compensatory or more efficient metabolic pathways in response to therapy.

Recent advances in cancer metabolism research have led to the development of a limited number of clinical trials, summarized in Table 1. Concurrently, Figure 7 highlights the most advanced metabolic-targeted therapies for GB. Many approaches are still under investigation in preclinical studies, and no significant clinical improvement has been achieved thus far. The efficacy of metabolic-targeted therapy relies on factors such as target metabolic pathway expression and cell plasticity. Cancer cells can utilize different metabolic pathways and adapt to changing environmental conditions or therapy, rendering a single metabolic pathway-targeting approach potentially insufficient. Furthermore, drug specificity and the use of targeted metabolic pathways by healthy cells, such as immune cells, must be taken into account, as inhibiting these pathways may cause undesirable side effects. While preclinical studies show promise, clinical trials are necessary to determine the safety and efficacy of these treatments for GB.

The intricate relationships between metabolic pathways, as well as the connections between metabolic and cell signaling pathways, warrant further investigation. Signaling pathways that modulate tumor metabolism have not been fully described in this review. For example, HIF-2α, a transcription factor activated in response to hypoxia, reprograms cellular metabolism and has been shown to maintain stemness [197]. Unfortunately, a phase II clinical trial showed no significant responses to HIF-2α inhibitors in recurrent GB patients (NCT03216499). Further studies are needed to fully evaluate HIF-2α inhibitors’ potential, especially in combination therapy. Autophagy, which is upregulated during stress conditions such as limited nutrient and oxygen availability, as well as in response to anticancer therapy, also requires further investigation. Although not a metabolic pathway in the strictest sense, autophagy is closely linked to cellular metabolism. As a catabolic process, it recycles cellular components, providing energy and building blocks for the cell, and confers a survival advantage to glioma cells in the hostile conditions of the tumor microenvironment. Autophagy can also regulate pro-growth signaling and metabolic rewiring of cancer cells, further supporting tumor growth. However, the use of autophagy inhibitors in GB treatment remains challenging [198]. Additional areas deserving further exploration include the role of the tumor microenvironment in shaping tumor metabolism, the crosstalk between cancer cells and stromal cells, and the underlying mechanisms of therapeutic resistance, particularly those related to metabolic-targeted therapies. Moreover, the involvement of epigenetic regulation and non-coding RNA regulators in GB metabolism presents another area deserving of deeper examination.

Despite numerous targeted therapies evaluated in clinical trials, the limited progress achieved in treating GB can be attributed to intratumoral heterogeneity, inadequate patient classification, and the crucial role of GSCs in therapy resistance and tumor relapse.

Novel therapeutic approaches based on the identification of metabolic vulnerabilities or the inhibition of multiple targets simultaneously may hold promise for improving treatment options for GB patients. Combination therapy, including metabolic therapies alongside radiotherapy, chemotherapy, immunotherapy, and targeted therapies, also shows potential in addressing GB’s therapeutic challenges. The challenges ahead include determining which metabolic pathways to target in each patient’s tumor, identifying pharmacological targets, and understanding how cancer cells modulate their metabolic strategy. To advance our understanding of GB and identify novel treatment strategies, continued investigation into the complex interplay between metabolic and molecular pathways, the tumor microenvironment, and therapeutic resistance is crucial.

## Figures and Tables

**Figure 2 ijms-24-09137-f002:**
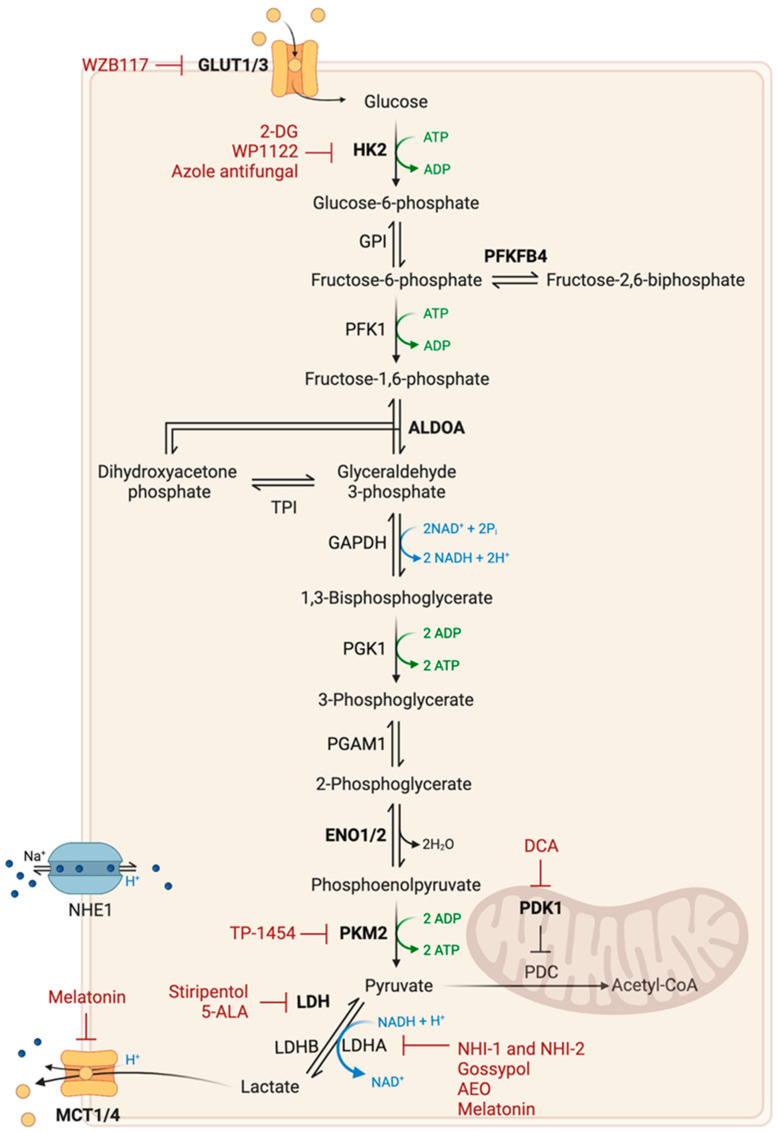
Warburg effect and related therapy. The Warburg effect, a hallmark of cancer metabolism, refers to metabolic reprogramming in cancer cells towards increased glucose uptake, glycolysis, and lactate production, even in the presence of oxygen. The most critical enzymes and transporters involved in this process are highlighted in bold and potential therapies in red. Targeting these enzymes and transporters is a potential strategy in cancer therapy. ADP: Adenosine Diphosphate; ATP: Adenosine Triphosphate; AEO: Anhydrous Enol-Oxaloacetate; 5-ALA: 5-Aminolevulinic Acid; 2-DG: 2-Deoxyglucose; ENO: Enolase; GAPDH: Glyceraldehyde-3-Phosphate Dehydrogenase; GLUT: Glucose Transporter; HK: Hexokinase; LDH: Lactate Dehydrogenase; MCT: Monocarboxylate Transporter; NAD^+^/NADH,H^+^ Oxidized/reduced Nicotinamide Adenine Dinucleotide; NHE1: Sodium-Hydrogen Exchanger 1; PGAM1: Phosphoglycerate Mutase 1; PFK1: Phosphofructokinase-1; PGK1: Phosphoglycerate Kinase 1; PKM2: Pyruvate Kinase M2; TPI: Triosephosphate Isomerase. Figure created with BioRender.com.

**Figure 3 ijms-24-09137-f003:**
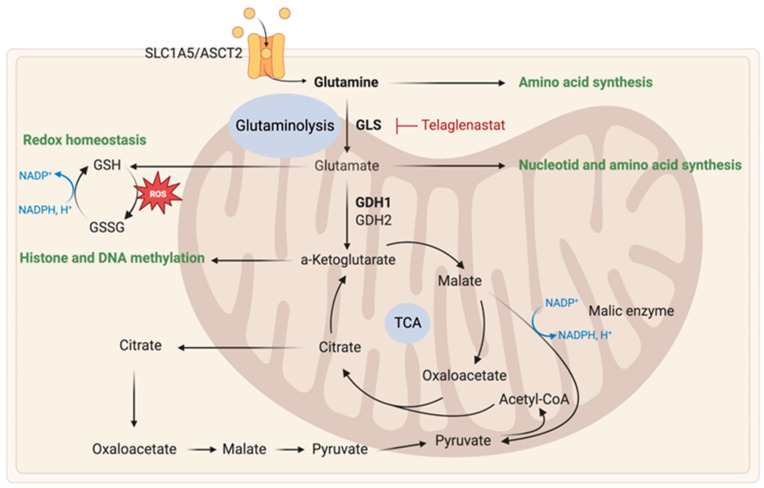
Glutamine metabolism pathways and processes in tumorigenesis and related therapy. Reprogrammed glutamine metabolism is involved in tumorigenic processes, which are highlighted in green. Reprogrammed glutamine metabolism supplies essential components for nucleotides, amino acids, and fatty acids biosynthesis, drives the TCA cycle, and maintains redox balance, as well as influencing epigenetic modifications through modulation of α-KG levels. Cancer cells take up glutamine via the transporter ASCT2/SLC1A5, which is then converted into glutamate by glutaminase. The conversion of glutamate to α-ketoglutarate by glutamate dehydrogenase fuels the TCA cycle. In GB, therapeutic targets are highlighted in bold and therapy in red. NADP^+^/NADPH,H^+^: Oxidized/reduced Nicotinamide Adenine Dinucleotide Phosphate; GLS: Glutaminase; GDH1/2: Glutamate Dehydrogenase 1/2; GSH/GSSG: Reduced/oxidized Glutathione; SLC1A5/ASCT2: Solute Carrier Family 1 Member 5/Alanine Serine Cysteine Transporter 2, member 5; TCA: Tricarboxylic Acid Cycle. Figure created with BioRender.com.

**Figure 4 ijms-24-09137-f004:**
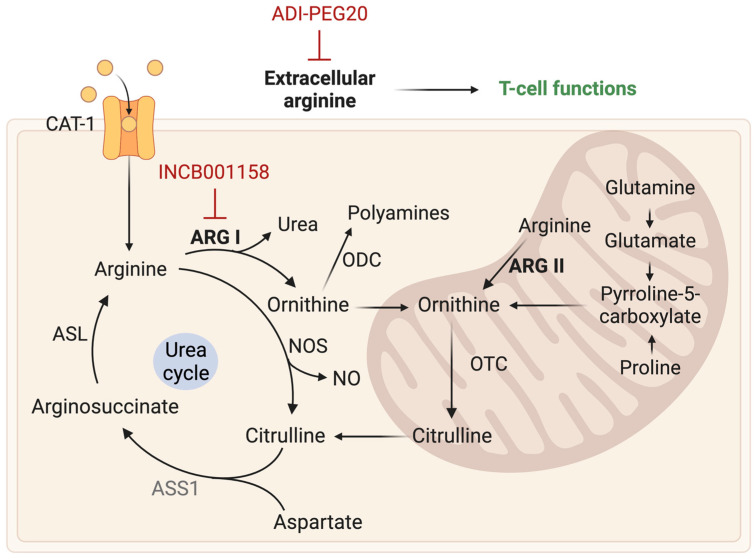
Arginine metabolism and related therapies. In glioblastoma, alterations in arginine metabolism are observed, characterized by an increase in arginine uptake, elevated arginase 2 (ARG II) levels, and a decrease in Argininosuccinate Synthase 1 (ASS1), an enzyme responsible for endogenous arginine synthesis. Therapeutic targets for GB are highlighted in bold, and therapies in red. The latter approach has shown promising results in ASS1-deficient glioblastomas. ADI-PEG20: Pegylated Arginine Deaminase; ASS1: Argininosuccinate Synthase 1; ARG: Arginase; ASL: Argininosuccinate Lyase; NO: Nitric Oxide; NOS: Nitric Oxide Synthase; ODC: Ornithine Decarboxylase; OTC: Ornithine Transcarbamylase. Figure created with BioRender.com.

**Figure 5 ijms-24-09137-f005:**
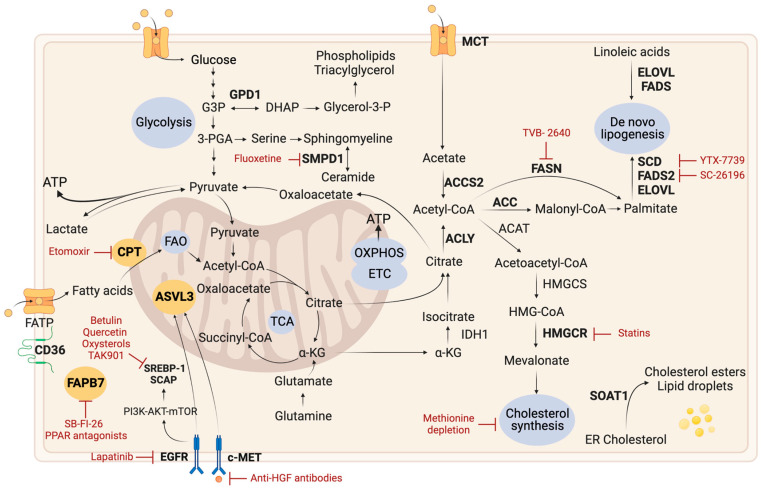
Lipid metabolism and associated therapies. In glioblastoma, alterations occur in multiple lipid biosynthesis and degradation pathways, including de novo lipogenesis, Fatty Acid Oxidation, cholesterol metabolism, and other lipid-related pathways. This figure highlights potential therapeutic targets in lipid metabolism that are highlighted in bold, as well as several therapies that target lipid-associated pathways, which are highlighted in red. ACC: Acetyl-CoA Carboxylase; ACCS2: Acetyl-CoA Synthetase 2; ACAT: Acetyl-CoA Acetyltransferase; α-KG: α-Ketoglutarate; ATP: Adenosine Triphosphate; CPT: Carnitine Palmitoyltransferase; DHAP: Dihydroxyacetone Phosphate; ETC: Electron Transport Chain; EGFR: Epidermal Growth Factor Receptor; ELOVL: Elongation of Very Long Chain Fatty Acids; ER: Endoplasmic Reticulum; FADs: Long-Chain Acyl-CoA Dehydrogenases; FAO: Fatty Acid Oxidation; FAPB7: Fatty Acid Binding Protein 7; FASN: Fatty Acid Synthase; FATP: Fatty Acid Transport Protein; G3P: Glyceraldehyde-3-Phosphate; GPD1: Glycerol-3-Phosphate Dehydrogenase 1; HGF: Hepatocyte Growth Factor; HMG-CoA: Hydroxymethylglutaryl-CoA; HMGCR: Hydroxymethylglutaryl-CoA Reductase; HMGCS: Hydroxymethylglutaryl-CoA Synthase; IDH1: Isocitrate Dehydrogenase 1; MCT: Monocarboxylate Transporters; OXPHOS: Oxidative Phosphorylation; P: Phosphate; PPAR: Peroxisome Proliferator-Activated Receptors; SCAP: SREBP-Cleavage-Activating Protein; SCD: Stearoyl-CoA Desaturase; SREBP-1: Sterol Regulatory Element Binding Proteins; SMPD1: Sphingomyelin Phosphodiesterase 1; SOAT1: Sterol O-Acyltransferase 1; TCA: Tricarboxylic Acid Cycle. Figure created with BioRender.com.

**Figure 6 ijms-24-09137-f006:**
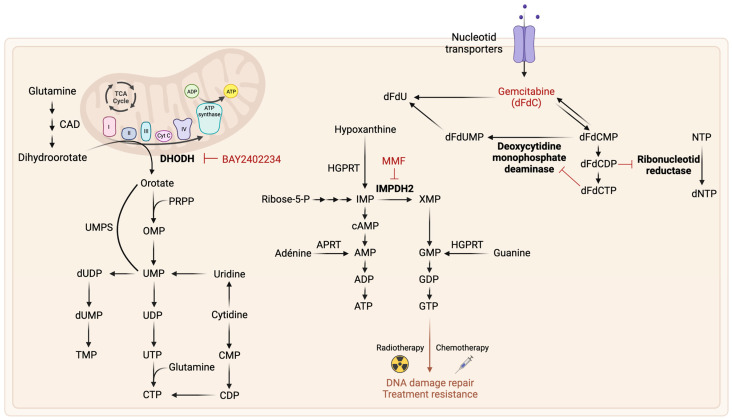
Nucleotide metabolism and associated therapies. This figure highlights potential therapeutic targets in nucleotide metabolism that are highlighted in bold, as well as several therapies that target nucleotide-associated pathways, which are highlighted in red. ADP: Adenosine Diphosphate; AMP: Adenosine Monophosphate; APRT: Adenosine Phosphoribosyltransferase; ATP: Adenosine Tri-phosphate; cAMP: Cyclic Adenosine Monophosphate; CAD: Carbamoyl-Phosphate Synthetase 2, Aspartate Transcarbamylase, and Dihydroorotase; CDP: Cytidine Diphosphate; CMP: Cytidine Monophosphate; CTP: Cytidine Triphosphate; DHODH: Dihydroorotate Dehydrogenase; DNA: Deoxyribonucleic Acid; dFdC: Difluoro-deoxy-cytidine; dFdCDP: Difluoro-deoxy-cytidine Diphosphate; dFdCMP: Difluoro-deoxy-cytidine Monophosphate; dFdCTP: Difluoro-deoxy-cytidine Triphosphate; dFdUMP: Difluoro-deoxy-uridine Monophosphate; dFdU: Difluoro-deoxy-uridine; dUDP: Deoxyuridine Diphosphate; dUMP: Deoxyuridine Monophosphate; GDP: Guanosine Diphosphate; GMP: Guanosine Monophosphate; GTP: Guanosine Triphosphate; HGPRT: Hypoxanthine-Guanine Phosphoribosyltransferase; IMP: Inosine Monophosphate; IMPDH2: Inosine Monophosphate Dehydrogenase 2; MMF: Mycophenolate mofetil; dNTP: Deoxynucleotide Triphosphate; NTP: Nucleotide Triphosphate; OMP: Orotidine Monophosphate; PRPP: Phosphoribosyl Pyrophosphate; TMP: Thymidine Monophosphate; UDP: Uridine Diphosphate; UMP: Uridine Monophosphate; UTP: Uridine Triphosphate; UMPS: Uridine Monophosphate Synthetase.

**Figure 7 ijms-24-09137-f007:**
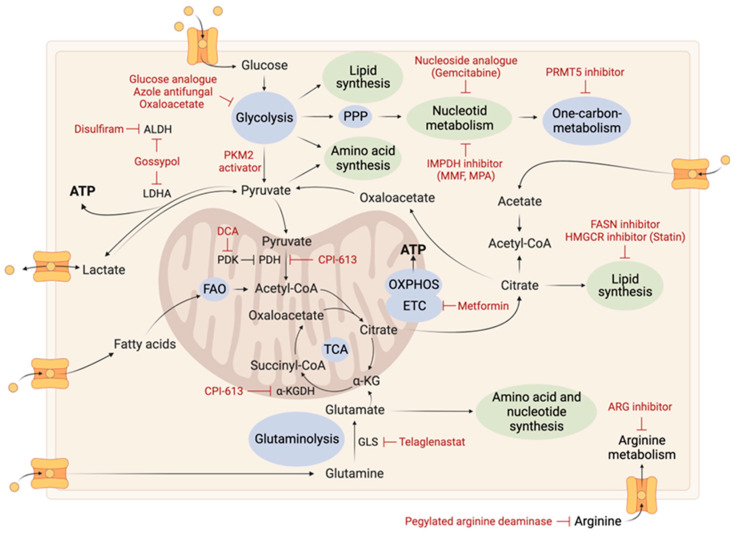
Advanced metabolic-targeted therapy in glioblastoma. α-KG: α-Ketoglutarate; ARG: Arginase; ATP: Adenosine Triphosphate; DCA: Dichloroacetate; ETC: Electron Transport Chain; FAO: Fatty Acid Oxidation; FASN: Fatty Acid Synthase; HMGCR: Hydroxymethylglutaryl-CoA Reductase; IMPDH: Inosine Monophosphate Dehydrogenase; MPA: Mycophenolic acid; MMF: Mycophenolate mofetil; OXPHOS: Oxidative Phosphorylation; PKM2: Pyruvate Kinase M2; PRMT5: Protein Arginine Methyltransferase 5; TCA: Tricarboxylic Acid Cycle. Figure created with BioRender.com.

**Table 1 ijms-24-09137-t001:** Main Clinical Trials in Metabolic Targeting for Glioblastoma Therapy.

Therapy	Mechanism	Trial Phase	Study Participants/Therapeutic Strategy	References
2-Deoxy-D-Glucose	Glucose analogue	I/II	Newly diagnosed GB patients treated with 2-DG in combination with radiotherapy	[30,31]
WP1122	Glucose analogue	I	Healthy volunteers	NCT05195723
Ketoconazole Posaconazole	HK2 inhibitor	I I	GB patients scheduled for radiotherapy	NCT04869449 NCT04825275
TP-1454	PKM2 activator	I	Patients with progressive solid tumors, treated with TP-1454 alone or in combination with immunotherapy	NCT04328740
Dichloroacetate	PDK1 inhibitor	I II	Recurrent GB Newly diagnosed and recurrent GB	NCT01111097 NCT00540176
Gossypol	Bcl-2 protein family and dehydrogenases inhibitor	I	Newly diagnosed GB treated in combination with TMZ with or without radiotherapy	NCT00390403
		II	Progressive or recurrent GB	NCT00540722
Anhydrous Enol-Oxaloacetate (AEO)	Oxaloacetate pro-drug	II	Newly diagnosed GB treated in combination with standard therapy	NCT04450160
Telaglenastat	GLS inhibitor	I/II	Advanced/metastatic solid tumors patients in combination with talazoparib (PARPs inhibitor)	NCT03875313
ADI-PEG20	Pegylated arginine deiminase	I	Recurrent high-grade gliomas in combination with chemotherapy	[76,77]
INCB001158	Arginase inhibitor	I/II	Advanced solid tumors in combination with chemotherapy	NCT03314935
TVB-2640	FASN inhibitor	II	Recurrent GB in combination with bevacizumab	NCT03032484
Atorvastatin	HMG-CoA reductase inhibitor	II	Newly diagnosed GB patients treated in combination with radiotherapy and temozolomide	NCT02029573
Gemcitabine	Nucleoside analogue	II	High-grade glioma patients treated in combination with radiotherapy	[123]
GSK3326595	PRMT5 inhibitor	I	GB	NCT02783300
CPI-613	TCA-targeted therapy	II	Solid tumor	NCT01832857
Metformin	AMPK activator, complex I respiratory chain inhibitor	II II	Recurrent or refractory GB treated with low TMZ plus metformin Newly diagnosed IDH wild-type GB patients with the OXPHOS+ signature in combination with standard therapy	NCT03243851 NCT04945148
Ascorbic Acid	Cofactor, antioxidant	II	Newly diagnosed GB treated in combination with the standard therapy	NCT02344355
Disulfiram	ALDH inhibitor	I/II	Patients with presumed GB treated with disulfiram and copper before surgery and during adjuvant chemoradiotherapy	NCT02715609

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
