# Peer review of "Glioblastoma Metabolism: Insights and Therapeutic Strategies"

_ijms, 2023, doi:10.3390/ijms24119137_

Round 1

Reviewer 1 Report

The manuscript by Chloé Bernhard et al., submitted in the journal IJMS, entitled “Glioblastoma Metabolism: Insights and Therapeutic Strategies". The objective of this review is to examine the metabolic changes in glioblastoma and investigate the role of specific metabolic processes in tumorigenesis, as as well as associated therapeutic approaches, with a particular focus on glioma stem cell populations.The manuscript has an importance in the scientific community as it deals with molecular pathways related to different treatments.

- The topics were adequately deepened, but what could be added are tables where the reader can relate the types of therapy with the metabolic pathway and the advantages/disadvantages.

- Indicate the limitations of the manuscript

- In the perspectives could be approached of new existing therapies that are in the clinical phase ( Trials) or in preclinical phase.

I congratulate the authors for this work.

English writing is adequate

Reviewer 2 Report

Bernhard et al, reported a comprehensive review of Glioblastoma Metabolism: Insights and Therapeutic Strategies. Overall, the authors have done a great effort and extensive work summarizing the metabolic processes underlying glioblastoma and the potential pharmaceutical targets, providing a detailed and well-organized overview of the relevant literature and the latest research done.

Despite this, I recommend some minor modifications and interconnections to improve the manuscript.

- Since one of the pivotal points of the review is the description of pharmaceutical targets, the authors may want to consider including a graphical abstract with the most relevant pharmaceutical approaches/ targets used so far. Even though the authors mentioned drugs use in the different graphical abstracts and paragraphs, they could also briefly summarize the different pharmacological targets or current therapeutics used or under investigation or approved by EMA or FDA, in a separate graphical abstract.

- Regarding cholesterol metabolism, the authors mentioned that in various in vitro and in vivo models, blocking cholesterol absorption or de novo biosynthesis has demonstrated anticancer effects (lines 455-456). Could the author support this part with proper references?

- Concerning cholesterol metabolism and specifically, the mevalonate pathway, it has been observed that FPP and GGPP, precursors of cholesterol in the de novo pathway, play a significant role in the prenylation of proteins involved in the development and progression of certain cancers. Authors may want to consider the role played by Ras, Rho protein in GB. (Lo HW. Targeting Ras-RAF-ERK and its interactive pathways as a novel therapy for malignant gliomas. Current cancer drug targets.) (Al-Koussa H, et al The Role of Rho GTPases in Motility and Invasion of Glioblastoma Cells. Anal Cell Pathol (Amst)) (Yufang Ma, Critical functions of RhoB in support of glioblastoma tumorigenesis, Neuro-Oncology).

- There are some effects between OXHPOS and cholesterol metabolism in glioma cell lines that authors may want to consider or comment on in the paper. FPP is also the precursor of Coenzyme Q (CoQ), an enzyme that has a critical function in the mitochondrial electron transport chain. An in vitro study demonstrated that the inhibition of CoQ biosynthesis with 4-nitrobenzoate (4-NB) significantly increased the cholesterol content in glioma cells, leading to a reduction in plasma membrane fluidity, decreased oxygen level, able to stabilise HIF-1α. As a result, this stabilization drove the metabolic switch to glycolysis in glioma cells. (Liparulo et al. Coenzyme Q biosynthesis inhibition induces HIF-1α stabilization and a metabolic switch toward glycolysis. The FEBS journal. ) Moreover, another study suggested that HIF-2α regulates the expression of genes associated with stemness and differentiation in GSCs. The study suggests that targeting HIF-2α could be a potential therapeutic strategy for glioblastoma. (Nusblat LM, et al. Gene silencing of HIF-2α disrupts glioblastoma stem cell phenotype. Cancer Drug Resist. 2020).

-          The authors mentioned throughout the paper the role and druggability of Mtor. Despite this, and taking into account the importance of mTOR pathway in GB, as in other cancer cell lines, authors should include more information about the rationale to use mTOR in GB treatment. I would also add this reference (Divé, Iet al. Inhibition of mTOR signalling protects human glioma cells from hypoxia-induced cell death in an autophagy-independent manner).

-          Although the authors separate in different paragraphs the various metabolic pathways druggable or used as a therapeutic strategy, the audience may lose the real take-home message of the review. Since there is a deep interplay among all these ramified and multifaceted metabolic pathways,  I would spend some words on how GB development, onset and progression, could lead to an overall metabolic remodelling and/or reprogramming and possible switch toward compensatory or more efficient metabolism, for the progression of the tumour.

- This is actually a caveat that authors should consider. The title of the paragraph “Metabolic interactions” could sound slightly misleading. Some readers could interpret it as an interplay between different metabolic pathways rather than cooperation between different cancer cells. I could rephrase it as “Metabolic interactions with surrounding cells” or “Metabolic interactions in tumour cells environment.”

-          If the authors are using Bio Render for creating the graphical abstracts, they should declare this in the manuscript.

 Minor editing of English language required. Authors should avoid some repetitions.

Round 2

Reviewer 2 Report

I am satisfied with the authors' revision and I hope that my suggestions and guidance have been appreciated.

There are no issues detected.